# Evidence-Based Recommendations for Herd Health Management of Porcine Post-Weaning Diarrhea

**DOI:** 10.3390/ani12141737

**Published:** 2022-07-06

**Authors:** Esben Østergaard Eriksen, Ken Steen Pedersen, Inge Larsen, Jens Peter Nielsen

**Affiliations:** 1Section for Production, Nutrition and Health, Department of Veterinary and Animal Sciences, University of Copenhagen, Grønnegårdsvej 2, 1870 Frederiksberg, Denmark; ken@sund.ku.dk (K.S.P.); inge@sund.ku.dk (I.L.); jpni@sund.ku.dk (J.P.N.); 2Ø-Vet A/S, Køberupvej 33, 4700 Næstved, Denmark

**Keywords:** post-weaning diarrhea, pig, decision-making, herd health management, decision-support tool, scientific summary, evidence-based veterinary medicine

## Abstract

**Simple Summary:**

In this paper, you will find recommendations on how to prevent post-weaning diarrhea in pigs kept in indoor pig herds. The recommendations are based on the scientific knowledge that is currently available. The authors first validated that Danish veterinarians working with pigs demanded such recommendations. Then, we collected papers written by other researchers who had summarized the scientific knowledge on different topics related to post-weaning diarrhea. From the papers, we extracted and synthesized 79 specific recommendations that may help veterinarians and pig producers make good decisions for their pig herd. The paper exemplifies a novel approach to summarizing and transferring science into practice that may be of interest to people that are not involved with pigs and post-weaning diarrhea.

**Abstract:**

Aided by their advising veterinarians, pig producers need to make difficult decisions regarding herd health management strategies. For instance, the preventive use of antimicrobials and medicinal zinc oxide must be substituted with more sustainable preventive approaches to porcine post-weaning diarrhea. Veterinarians and pig producers may find assistance in knowledge based on evidence in this regard; however, the overwhelming scientific literature is not always readily available. The overall aim of this paper is to suggest herd health management decision-support tools that can aid veterinary-assisted decision making in the control of porcine post-weaning diarrhea at a tactical level. The first objective was to validate the need for a herd health management concept, including two decision-support tools. The second objective was to develop evidence-based recommendations that can aid veterinary-assisted decision-making for the herd health management of post-weaning diarrhea. The first objective was investigated by a questionnaire-based study among veterinary pig practitioners in Denmark. For the second objective, we conducted a scientific summary based on scientific review papers identified through a systematic search in three databases. From the papers, we synthesized and extracted 79 specific recommendations. In this paper, we report comprehensive evidence-based recommendations for the herd health management of post-weaning diarrhea.

## 1. Introduction

Porcine post-weaning diarrhea (PWD) is a multifactorial condition, characterized by the clinical sign of diarrhea in the first 14 days after weaning, commonly occurring in the intensive indoor swine production (see [1] for a complete definition). In Denmark and other European Countries, PWD has frequently been prevented by adding high doses of medicinal zinc to the weaner diet [2,3,4]. 

However, the preventive veterinary use of zinc will be prohibited in the European Union no later than June 2022 [5] due to environmental concerns and co-selection for antimicrobial resistance [6]. Thus, European swine producers are undergoing a major transition to zinc-free weaning. If no other preventive measures are installed, the incidence of post-weaning diarrhea and the antimicrobial consumption is expected to increase. 

Fortunately, there is an overwhelming amount of scientific evidence regarding post weaning diarrhea and its prevention (e.g., [7,8,9]). This forms a sound basis for the prevention of PWD through herd health management strategies. Yet, to do any good, the evidence must be disseminated and implemented in the swine production, and this task is not straightforward. The animal behavior science pioneer Dr. Temple Grandin captured it this way: “*I have learned that successful transfer of knowledge and technology to industry often requires more work than doing the research*” [10]. Even when science-based information is disseminated and the benefits of doing so appear obvious, livestock producers can still be reluctant to adopt new herd health practices [11]. The study of farmers’ decision-making regarding herd health gives multiple explanations of why [11,12]. Evidence-based decision-support tools can bridge the gap and bring the scientific knowledge—for instance regarding PWD—into practice.

Decision-support tools lead livestock producers through clear decision stages and/or present the likelihood of different outcomes when a given alternative/solution is selected [13]. The format of decision support tools can vary from simple calculators, fact sheets or decision-trees to complex monitoring and guidance systems. They can be both digital (e.g., online or software-based) or paper-based. There is considerable variation in their uptake and impact, and emphasis should be put on the selection of format and the design [13]. 

Herd health issues, such as PWD in pigs, are often complex and have multifactorial causation. Consequently, the relevance of different alternatives will differ between herds and priorities of the pig producer. Furthermore, a study of the adoption of best herd health practices by sheep farmers suggested that non-compliance should not be viewed as one type of behavior; non-compliance behaviors are heterogenic and have different explanations, and therefore one-size-fits-all solutions will likely be ineffective in changing them [14]. We assume this is also true for pig producers. A solution to these two problems is to tailor-make herd health strategies for the herd-specific context and manager. 

Tailor-made advice is offered to pig producers through veterinary herd health consultancy, and veterinarians are a trusted source of information, especially regarding herd health decisions. Additionally, recommendations by peers and advisors, including veterinarians, is an efficient way of disseminating decision-support tools to farmers [15]. Therefore, we wanted to design decision-support tools for the prevention of PWD to be used in the context of a veterinary herd health advisory situation. We further specified that it should be used in the context of intensive indoor pig production and decided that the tools should be used for decisions at a tactical level (as reviewed by Gray et al. [16]). Thus, recommending decisions at a strategic level, e.g., investing in new barn-systems or building additional facilities were precluded. 

We also specified the outcomes the end-user may desire and expect when using the tools. We chose to develop tools focused on limiting the incidence of clinical diarrhea in the first 14 days after weaning. Accordingly, the end-user might be led to decisions that are costly, limiting the productivity or have other unwarranted consequences. It will be up to the pig-producers and their advisors to weigh cost and benefits. Finally, we limited our focus to management-related decisions, and we did not include the feed composition and feed and water additives in our work. However, we did consider the methods by which feed and water are provided.

Within the framework defined above, we conceptualized a full herd health management concept for herd health management of PWD, including two decision support tools: evidence-based recommendations and a questionnaire for herd-level risk assessment inspired by those previously developed to audit and aid decisions regarding biosecurity in pig herd (e.g., [17,18]). To ensure their uptake, decisions-support tools in agriculture must match the actual problems, demands and working patterns of the end-user [13,15]. Therefore, the demand for the envisaged decision support-tools should ideally be validated before they are designed.

The overall aim of this paper is to suggest herd health management decision-support tools that can aid veterinary-assisted decision-making on the control of porcine post-weaning diarrhea at a tactical level.

Our first objective was to validate the need for our herd health management concept for PWD, including the two envisaged decision-support tools. Our second objective was to develop evidence-based recommendations that can aid veterinary-assisted decision-making on herd health management of PWD.

## 2. Materials and Methods

### 2.1. Study 1: Validation of Demand

Study 1 was a validation of the need for our herd health management concept for PWD among Danish veterinarians. We developed a short questionnaire guided by Stone [19]. The respondents were asked to provide the demographic characteristics described in a previous study [20]. Then, they were presented a herd health management concept for the removal of medicinal zinc usage and asked to what extent they were interested in the concept and whether they thought it would create value for pig producers and veterinarians (answers not reported). 

Finally, we asked the respondents to rate how useful they perceived the five different sub-elements of the concept when providing herd health consultancy regarding medicinal zinc removal and PWD. Finally, we asked whether the respondents would like to receive the final concept when it was developed, and if so, we requested their e-mail addresses. We disseminated the questionnaire to all Danish veterinary pig practitioners who were responsible for the herd health consultancy in at least one weaner pig herd as of 31 August 2020. The study population (n = 112) and their e-mail addresses (n = 90) were obtained as previously described [20]. Questionnaires were sent out electronically using SurveyXact (Rambøll Management Consulting A/S) as of 14 September 2020, and a reminder e-mail was sent to non-responders 21 September 2020. The data was interpreted based on descriptive statistics. 

### 2.2. Study 2: Developing Evidence-Based Recommendations

The methodological approach for the creation of evidence-based recommendations is outlined in Figure 1, and we describe it in detail below.

Dicks et al. emphasized the importance of incorporating well-synthesized science into decision-support tools. This is achieved by using scientific summaries based on (preferably systematic) reviews as the knowledge base, rather than a few studies, datasets or expert opinions [21]. Plentiful reviews covering different aspects of PWD prevention were available. Therefore, we choose to make a scientific summary based on published peer-reviewed review papers as a foundation for evidence-based recommendations. 

First, the authors of the present paper created a list of items believed to influence the PWD incidence. The list of items was circulated among seven Danish researchers working with post-weaning diarrhea, and these researchers provided comments and additional items to the list. To obtain review papers covering all items, we conducted a systematic literature search 5 May 2021 in the databases Web of Science, Cab Abstract and PubMed for papers published from year 2000 and onwards. In Web of Science, we searched in Keywords Plus^®^, in CAB abstracts we searched in the abstract, title, original title, broad terms, heading words, identifiers and CABICODES, and in PubMed, we searched in MeSH terms. 

We used two search blocks, here exemplified in Web of Science terminology: (KP = (pig$ OR swine$ OR piglet$ OR weaner$ OR porcine)) AND (KP = (“post wean*” OR “post-wean*” OR wean* OR diarrh* OR enteritis OR scour$)). We either used the document type functions in the databases or added the block (review OR meta-analy*) to restrict the results to review papers, and the search was restricted to documents in the English language. The papers were sorted as outlined in Figure 2 (PRISMA flowchart figure). 

Thematically, reviews of the association and causation between a given item and the occurrence of PWD was given primary priority, and secondarily, reviews of other health- and/or performance-related outcomes, such as growth performance or feed intake, as well as measures of gut health, e.g., enteric histological, immunological and microbiological measures, were included. 

Only the immediate post-weaning period (approximately 14 days) were of interest. In reviews discussing control measures of infectious agents, we mainly considered the evidence related to enterotoxigenic *E. coli*, *Salmonella* species and rotavirus, as we regarded these as the most relevant in the causation of PWD. Reviews that were recently published and/or systematic (or at least with a description of a sound methodology) were prioritized over older narrative reviews. 

Comprehensiveness was also prioritized. That is, full papers with a narrow focus or papers including longer paragraphs that cited more sources regarding a given item, were prioritized over shorter paragraphs with few cited sources; such paragraphs often occurred in reviews with a broad scope. The search in the three databases yielded 1057 papers after removal of duplicates, and the selection of reviews is described in Figure 2 (modified from [22]). 

After screening and sorting the papers, 82 were consider relevant for the scientific summary, of which 37 were cited in the recommendations. If an item was not sufficiently covered by a review detected in the systematic search, additional searches were conducted in the Web of Science database. Here, we first prioritized to identify relevant review papers, and in their absence, we looked for original research studies of the item and its effect on PWD and other relevant parameters. This led to the inclusion of an additional 10 papers and a book chapter.

## 3. Results and Discussion

### 3.1. Study 1: Validation of Demand

The questionnaire was completed by 37 (41%) of the veterinarians. The demography resembled one observed in a previous a study with the same target population [20], thus, indicating an absence of selection bias. Of the 37 responding veterinarians, 33 (89%) requested the material and provided their e-mail address. We interpreted this as a sign of true (rather than just “stated”) interest in the material from the vast majority of the pig health practitioners.

The results of Study 1, as displayed in Figure 3, confirmed that the target group, the veterinary pig practitioners, demanded the envisaged herd health concept. However, the checklist for a herd audit was the least demanded tool, as only 32% of the veterinarians deemed it to be useful. We had considered to develop this (in the format of a questionnaire producing auto-generated reports) in Study 2; however, our initial indications of the tool indicated that this would overcomplicate the dissemination of the knowledge. Taken together, we expected a low uptake and did not complete and report on the construction of this tool. 

Study 1 also showed a demand for tool for quick, easy and precise assessment of the prevalence of diarrhea and for an approach to effect evaluation of the zinc-removal and new initiatives. Accordingly, we are currently developing such a methodology on an empirical basis. Conclusively, there was a strong demand for a complete overview of the possible preventive measure and risk factors for PWD. Accordingly, we conducted Study 2, and we provide the resulting evidence-based recommendations in the latter part of this paper.

### 3.2. Study 2: Evidence-Based Recommendations

The objective for Study 2 was to propose evidence-based recommendations on prevention of PWD as a decision-support tool that could aid veterinary-assisted decision-making for herd health management of porcine post-weaning diarrhea. The synthesis of the scientific evidence resulted in 79 specific recommendations covering 37 different items. 

Each item is listed in Table 1, Table 2, Table 3, Table 4, Table 5, Table 6 and Table 7 with a brief summary of the current evidence, the specific recommendations and citations of the included scientific papers. The items are divided into the thematic sub-headings: genetics (Table 1), management before weaning (Table 2), management during weaning (Table 3), management after weaning (Table 4), feeding and water strategy (Table 5), biosecurity and biomanagement (Table 6) and management of specific pathogens (Table 7). We included a Danish translation of the table contents as Appendix A.

### 3.3. Limitations and Perspectives

In this paragraph, we consider the strengths and limitations of our methodology, briefly introduce the importance of feed composition and additives, and finally we discuss how recommendations may be implemented in practice. 

#### 3.3.1. Methodology 

The methodology applied in this paper (see Figure 1) is, to our knowledge, novel. The ideas described by Dick and colleagues and colleagues [21] were fundamental for the conceptualization of methodology. Evidence-based recommendations are clearly limited by the amount of available evidence. While some recommendations are backed by reviews, finding good evidence of a clear effect on the PWD incidence (e.g., recommendation 33 regarding the room temperature) others are based on certain assumptions (e.g., recommendations 16–20 regarding handling during weaning) or reviews of a few studies indicating an effect on performance or paraclinical measures rather the incidence of PWD (e.g., recommendation 42 regarding feeding on the floor). Hence, users should pay attention to the level of evidence described in columns “What do we know?” (Table 1, Table 2, Table 3, Table 4, Table 5, Table 6 and Table 7). 

Recent reviews were not always available. In these cases, the methodology will likely miss the most recently published evidence. Another limitation was that narrative reviews dominated, and systematic reviews and meta-analyses were rare. Hence, the study selection and interpretation in the reviews may be biased, and consequently our recommendations may be so. An additional bias may be introduced as we were often forced to synthesize recommendations from the reviews, and thus the recommendations may suffer from our biased reading of the papers or limited ability to understand the subjects. 

Reviews regarding the intestinal health of post-weaning pigs will likely also be published at a high rate in the future. Based on our experience, we plea for reviews applying a systematic approach, to preferably meta-analyze the results if applicable, and that the authors, based on their conclusions, dare to provide a set of clear, practically relevant recommendations that can be directly harvested for purposes like the present review. A commonly used method for the present objective would be to base the recommendations on the opinions of an expert panel. Compared to this approach, we suggest that extracting recommendations from review papers will be less prone to bias. 

Hence, we believe that the novel methodology presented in this paper was an efficient way of obtaining reasonable evidence-based recommendations, and we propose that it may be reused for similar objectives. It is also important to validate the demand before conducting the work [13] similar to what we did in Study 1. Veterinarians facing any herd-health problems can use a simplified approach: create a list of items together with colleagues, identify one or a few review papers so that all items are covered and extract the recommendations.

#### 3.3.2. Feed Composition and Additives 

As previously defined, we did not aim to summarize the scientific evidence about feed composition and feed and water additives. However, this is a cornerstone in the herd health management of PWD, and herd audits should not only evaluate the compliance to the recommendations provided in Table 1, Table 2, Table 3, Table 4, Table 5, Table 6 and Table 7 but also the feed composition and additives. This will often require the expertise of a feed advisor. Our systematic sorting of the literature resulted in 155 review papers published after the year 2000 that discussed at least one item related to feed composition and/or additives; thus, clearly, there is also an overwhelming amount of evidence to bring into practices in this regard. We highlight three recent reviews [36,68,69] that comprehensively reviewed feed composition and additives as measure to prevent PWD without zinc and antibiotics. Additionally, Pluske, Turpin and Kim exemplarily provided a selected set of specific recommendations at the end of their review paper from 2018 [8]. Below, we briefly mention some of the topics and recommendations discussed in the literature. 

The crude protein level should be reduced while observing that eventual effects on growth rate is acceptable and protein from legumes, including soy beans, should be avoided or fermented, while animal protein sources was considered favorable [8,36,68,69,70,71,72]. 

Adequate and balanced supplies of amino acids are important for growth performance, especially when reducing the crude protein level as well as to support the health and development of the intestine and immune system, and specific recommendations for the amino acid levels can be extracted from the reviews [8,70,73,74,75]. 

Early-weaned pigs are not good at digesting fat, and thus dietary fat should be easy to digest. Short and unsaturated fatty acids are easier to digest than long and saturated fatty acids, and the position of the fatty acids in triglyceride structures plays a role. Therefore, it was recommended to maximize the unsaturated:saturated fatty acids ratio, to maximize the ratio of n-3:n-6 polyunsaturated fatty acids and to include n-3 polyunsaturated fatty acids and medium chain fatty acids in the diet [36,76,77]. 

Some feed components, most notably soy beans, contain anti-nutritional factors that should be avoided [55,78,79]. The minerals added to the diet must be considered, and calcium is given special attention due to its effect of the acid-binding capacity of the feed [8,36]. 

The importance of the composition of carbohydrates was also extensively discussed in multiple reviews, e.g., [36,68,69]. Among other things, adding fermentable fibers may induce a shift in the microbial composition in the colon towards fiber-fermenter bacteria and away from unwanted protein-fermenting bacteria [36,68,69]. Thus, there might be an overlap between the composition and additives, in the sense that some fiber-rich feed components may be termed prebiotics due to this modifying ability [69]. 

When we consulted our expert panel and the literature reviews, numerous commercially available feed additives appeared: antioxidants including vitamin E, antisecretory factors, bioactive compounds from algae or seaweed, blood plasma, bovine colostrum, chitosan, clay minerals, enzymes (e.g., phytase or xylanase), essential oils, high-intensity sweeteners, lactose, nucleotides, organic acids, phytobiotics, prebiotics (e.g., inulin or different oligosaccharides), probiotics (e.g., *Lactobacillus* spp. *Enterococcus* spp., *Bacillus* spp. *Clostridia* spp. or yeasts), synbiotics, tannins and vitamins. Water additives included electrolytes and organic acids. 

It is difficult to give universal recommendations regarding feed additives. We suggest to critically assess the producer’s documentation and consult a relevant literature review before considering a feed additive. After implementation, the effect should be evaluated in the specific herd as described below in the last part of the final paragraph of our discussion.

#### 3.3.3. Implementation in Practice

The ultimate purpose of the present work was to bring scientific evidence into practice. Now that a set of recommendations is available, how is implementation in practice achieved? As we argued in the introduction, veterinarians are often trusted advisors in the herd health decision-making. Studies have demonstrated how tailor-made advice can be provided through veterinary consultancy and result in high compliance with recommendations leading to actual reductions in the antimicrobial use in pig herds [80,81].

In such a process, the veterinarian must first conduct a herd audit and select measures to recommend in a given herd. In this regard, we suggest considering causation. The effect of the measures recommended in Table 1, Table 2, Table 3, Table 4, Table 5, Table 6 and Table 7 on the incidence of PWD are commonly mediated through larger pathways, i.e., by reducing weaning associated stress, minimizing the presence of pathogens, or by enhancing the resilience of the pigs. Thus, the selection of recommendations should rely on an expert evaluation of which part of the causation of PWD where there is the greatest room for improvements in the given herd. For example, the demonstration of a large burden of enterotoxigenic *E. coli* indicates a special relevance of recommendations lowering the pathogen burden (e.g., recommendations 57–68) or increasing the resistance towards the pathogen (e.g., recommendations 76–78). 

The next obstacle may be to convince the pig producer that the recommendations should be implemented and facilitate that they are actually implemented. To secure success in this difficult task, we recommend the review by Ritter and colleagues to all veterinary pig practitioners. This paper gives an overview of the determinants of farmers’ compliance when herd health management advice is provided [12]. 

The final part of a herd health management process entails an evaluation of the effect of the implemented measures. This is crucial, since herd health problems are multifactorial and complex, and the effect of new initiatives should be expected to be herd-specific [82]. We consider herd-specific trials as the highest standard of evaluation in herd health management. This entails selecting at least one measure of effect (e.g., the occurrence of diarrhea, antibiotic use or daily weight gain), establish a way to monitor it and compare the data collected in (groups of) animals with and without the initiative under evaluation [83]. The golden standard is a fully randomized design with parallel groups, i.e., some pigs are randomly assigned to receive the new initiative while some are not receiving it, and the performance of the two groups is compared [83]. However, this is often impractical, and a “before–after” design, where data collected before and after the given initiative is implemented is compared, can be a good alternative; however, fluctuations of confounding factors over time must be considered. For this purpose, methods for continuous implementation and the evaluation of new initiatives in livestock productions have been described [82,84].

## 4. Conclusions

In conclusion, this paper presented comprehensive evidence-based recommendations for the prevention of post-weaning diarrhea.

## Figures and Tables

**Figure 1 animals-12-01737-f001:**

Outline of how we made our evidence-based recommendations.

**Figure 2 animals-12-01737-f002:**
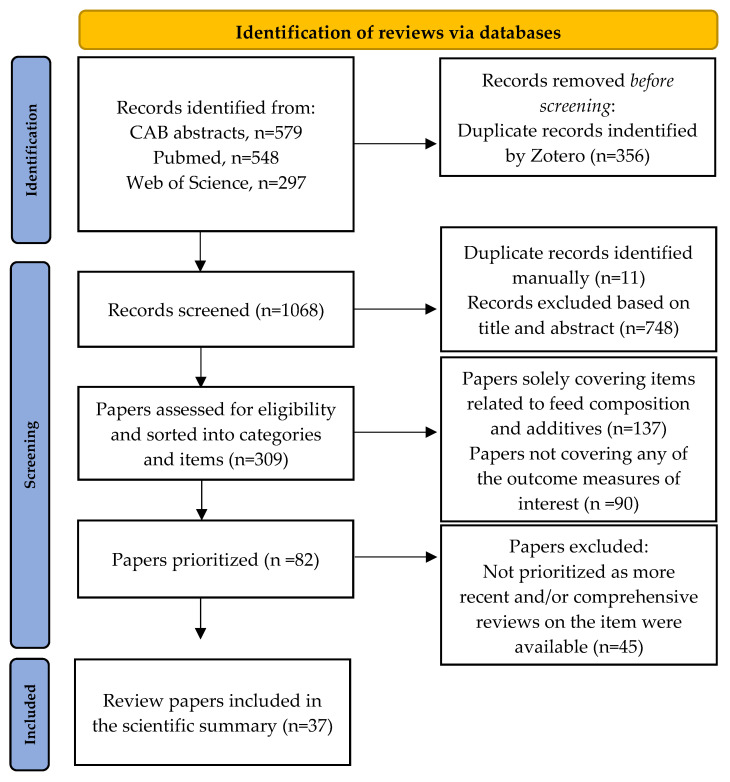
Flow chart of the searching, sorting and inclusion of review papers for the scientific summary. Modified from [22]: Page MJ, McKenzie JE, Bossuyt PM, Boutron I, Hoffmann TC, Mulrow CD, et al. The PRISMA 2020 statement: an updated guideline for reporting systematic reviews. BMJ 2021; 372: n71. https://doi.org/10.1136/bmj.n71 (accessed on 8 June 2022).

**Figure 3 animals-12-01737-f003:**
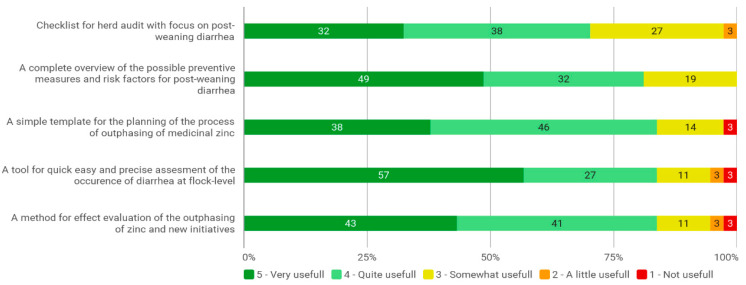
Danish veterinarians scoring the usefulness of five different tools for herd health advice regarding the out-phasing of medicinal zinc.

**Table 1 animals-12-01737-t001:** Genetics.

Item	What Do We Know?	Recommendations	Sources
Breed	The breed is known to cause differences in the intestinal microbiota in pigs housed in the same environment. Comparisons between breeds of the PWD incidence and gut health after weaning were not mentioned in reviews, and original research articles was not found in an additional search.	No specific recommendations are given on this item.	[23]
*E. coli* resistant animals	Candidate genes encoding the susceptibility to adhesion to the intestinal epithelium by the F4 and F18 fimbria-types has been identified, and it is possible to breed pigs resistant to specific fimbria types. For instance, a Danish program bred towards an allele of the MUC4 gene associated with resistance against certain ETEC F4 variants. A major limitation is that new *E. coli* strains with altered adherence mechanisms will likely evolve and successfully proliferate in resistant pigs/breeds. None of the reviews reported breeds that had generalized resistance to *E. coli*-associated PWD.	No specific recommendations are given on this item.	[9,24]

**Table 2 animals-12-01737-t002:** Management before weaning.

Item	What Do We Know?	Recommendations	Sources
Age at weaning	Early weaning might have long-term consequences for gut development, health and performance. Weaning before 21 days of age is prohibited in the European Union, and it is not advisable based on the available evidence. Intestinal morphology is more severely affected by weaning before 28 days of age, and the growth check and reduced feed intake after weaning is more prominent when weaning pigs before 28 days of age. The reviews generally recommended to avoid weaning before 26–30 days of age. It might be beneficial to increase the weaning age even further; however, the evidence supporting this is ambiguous.	Wean pigs no earlier than 28 days of age. Increasing the weaning age further can be considered.	[9,23,25,26,27,28,29,30]
Body weight at weaning	The body weight at weaning predicts the long-term performance of the individual pig. The weight at weaning can be condensed as a function of litter size, birth weight, pre-weaning management, genetic disposition and weaning age. Thus, the weight at weaning can be viewed a proxy for specific protective/risk factors for PWD, such as colostrum intake or age at weaning. In this scientific summary, we have chosen to give the recommendations specifically for the items related to PWD for which weight at weaning is a proxy, not on how to enhance weaning weight. Recommendations on how to increase the weaning weight is available in the cited source. Nevertheless, weight at weaning can be relevant to monitor, in order indirectly measure the other pre-weaning management items and age at weaning.	No specific recommendations are given on this item.	[31]
Colostrum intake	Colostrum intake is important for the post-natal intestinal development. The role of colostral immunity in PWD is not well described in the literature. However, low colostrum intake increases the risk of other post-weaning outcomes, such as mortality and post-weaning multisystemic wasting syndrome. Split-suckling, may help to ensure adequate colostrum intake for all piglets within a litter.	2. Practice split suckling in litters with piglets at risk of receiving too little, e.g., small piglets and piglets with the intrauterine growth restriction syndrome (IUGR) born in large litters. 3. Provide split-suckling as early after farrowing as possible, no later than 24 h post-farrowing, suggestively in 2 h bouts. 4. Weak piglets might benefit from assistance to find a teat and suckle. 5. Weak piglets might benefit from colostrum supplementation from milked sows.	[9,23,26,31,32,33,34,35,36]
Birth weight	Low birth weight, often associated to the intrauterine growth restriction syndrome (IUGR), is a risk factor for low colostrum intake, pre-weaning death, low weaning weight and other negative pre-weaning outcomes. IUGR piglets have a slower gut development the first week of life and a different intestinal microbiota. The review articles did not aim to scrutinize post-weaning outcomes thoroughly; however, we identified original research studies reporting low birth weight/IUGR to be associated to reduced post-weaning growth rate and impaired inflammatory response, absorptive capability and antioxidative capability in the intestine. One study reported an increased hazard of PWD, while two other studies presented equivocal results on the mean diarrhea score. The proportion pigs that are born with low birth weight can be affected, e.g., by feeding of the sow during gestation or by selecting less prolific sows or genetic linages/breeds.	6. Minimize the proportion of pigs born with low birth weight and intra uterine growth restriction syndrome.	[1,30,31,37,38,39,40]
Creep-feeding	Creep feeding supposedly prevents PWD by familiarizing the digestive apparatus with solid feed, i.e., stimulating production of enzymes and tolerance to feedstuff antigens. How and whether creep feed provision modulates the gut microbiota is unclear. However, the evidence of the effect of creep feed on PWD incidence is ambiguous. To be efficient, creep feed must not only be provided; it must also be eaten, preferably in large amounts. One source suggested that pigs should preferably not be weaned before they have consumed approximately 600 g of creep-feed, which however, will require a dramatic increase in weaning age under most production schemes. Later weaning allows for increased creep feed consumption before weaning. At common weaning ages (e.g., 28 days), not all pigs will eat creep-feed yet, and those that do often eat small amounts. It is not clear when creep feed should be provided, and contradictory recommendations were given. While some reported it to be sufficient to provide creep feeding 2–3 days immediately before weaning to satisfactory gain the benefits, others recommended longer periods of creep feed. Nevertheless, almost no pigs consume creep-feed before 14 days after birth, and consumption is limited until 21 days of age. The way creep feed is provided, the composition and form of the feed have a role in the amount of creep-feed ingestion.	7. Provide creep feed with a target of at least 600 g consumed feed per piglet before weaning. 8. Focus on maximizing the proportion of pigs that ingest creep-feed and the amount of feed eaten. 9. Creep-feed provision before day 14 after birth, might not have any effect. 10. Creep-feed provision before day 21 after birth likely has a minor effect. 11. Liquid/gruel feed is recommended. 12. If pellets are used, use soft/large pellets. 13. The palatability and complexity of the diet should be high, while the nutrient density should be moderate. 14. Use “play feeders” and/or feeders that are easy to localize and access and allow social feeding behaviors.	[7,8,23,26,31,34,36,41]
Supplemental feeding with milk replacer	Supplemental milk feeding in the pre-weaning period was cited to increase survival of suckling piglets and the pre-weaning weight gain, at least for low birth weight pigs. One study was cited to document increased post-weaning weight gain, and one study was cited to show reduced pre-weaning diarrhea incidence and reduced post-weaning diarrhea incidence. However, sow milk should remain the primary source of nutrients for suckling pigs. Supplemental milk is primarily consumed by small piglets staying with sows with poor milk production and/or large litters. Differences in the efficiency of milk supplementation between producers will likely occur. It is unclear whether the addition of functional ingredients to the supplemental milk has beneficial effects.	15. Provide supplemental feeding with replacement milk in the pre-weaning period for sows with large litters and/or poor milk production.	[26,31,36]
Vaccination of sows	Vaccinating sows prior to farrowing stimulates the production of specific antibodies that offers piglets protection through the colostrum and milk. Different types of ETEC-vaccines are available from a number of producers. None of the included reviews discussed the impact of the maternal immunity on PWD or post-weaning health and performance in general.	No specific recommendations are given on this item.	[32,42]
Antimicrobial treatments in the pre-weaning period	Antimicrobial treatments in the pre-weaning period impact the intestinal microbiota. The effect is traceable for at least five weeks, i.e., into the post-weaning period. Immunological processes, and the digestion and absorption of nutrients is also affected. If and how this influence post-weaning performance and clinical outcomes was not mentioned, and a rapid search for additional literature did not reveal studies measuring such outcomes.	No specific recommendations are given on this item.	[23]
Partial weaning of litters	None of the reviews touched upon the practices where piglets are weaned in two stages, providing the smallest piglets with some exclusive additional time with the sow. No relevant original research papers could be identified.	No specific recommendations are given on this item.	

**Table 3 animals-12-01737-t003:** Management during weaning.

Item	What Do We Know?	Recommendations	Sources
Gradual weaning	Sow-controlled and multi-suckling housing systems have been developed to allow a more gradual weaning in the intensive production. This summary focus on changes within existing systems; see the published reviews on the item if this major production transition is considered. The management practice “intermittent suckling” is another way to gain a more gradual weaning process. This entails intermittent separation of piglets and the sow in the final stage of the planned lactation. This will increase solid feed consumption before and after weaning, increase post-weaning growth and positively affect the intestinal morphology after weaning. However, intermittent suckling do not increase the proportion creep feed “eaters” but dramatically increases the amount of feed consumed by the “eaters”. An effect irrelevant for PWD prevention, yet noteworthy, is that three days of overnight separation (16 h) in the fourth week of lactation will induce estrus and possibly mating during lactation with a commercially acceptable rate. Disadvantages of intermittent suckling are that it requires additional labor, and a drop in growth rate may be seen in the late lactation. The latter will, however, be compensated by better performance in the first week after weaning. The reviews did not agree on whether the practice caused additional stress to the sow and piglets or not; however, the review suggesting it causes stress to the sows, did not cite any literature. The direct effect on PWD was not reported. Separating the piglets from the sow at a (much) older age (e.g., 6–8 weeks) will also indirectly lead to a more gradual weaning. In older studies, when this weaning age was common practice in indoor productions, 50–80% of the consumed energy just before weaning was creep feed. Currently, when piglets are commonly weaned at three-four weeks of age, the piglets consume little creep feed and rely almost exclusively on milk at the time of separation from the sow. In conclusion, the possible recommendations will generally require strategic decisions (e.g., build new barn facilities or systems). This includes the recommendation to practice intermittent suckling in combination with lactation periods >33 days, which however might be manageable in some instances within existing systems.	No specific recommendations are given on this item.	[30,34,43]
Handling during weaning	It is well established that a weaning-induced stress response may impair the immune system and gut function and thus enhance the susceptibility to PWD. Moving the piglets from the farrowing pen into nursery pens or the transport vehicle is hypothetically a source of stress, and the impact of the procedure would likely depend on how it is performed. However, no review or single studies of the effect of handling during movement of weaned pigs were identified. In finisher pigs, effects are well-documented, and a substantial amount of evidence on how to move pigs in the easiest and least stressful ways is available. We deemed it to be fair to critically generalize the recommendations to weaned pigs.	16. The distance that pigs are moved/herded should be as short as possible. 17. Avoid mixing with unfamiliar animals or at least only mix pigs that are going to stay together in the nursery. 18. Move pigs in smalls groups, e.g., single litters. 19. Ensure that alleys are well enlightened and that pigs are not forced to move from light to dark rooms. 20. Avoid high noise levels in the alleys that the pigs need to walk through.	[28,30,44,45]
Transport	The literature is dominated by studies of older pigs, and this was considered in addition to the limited evidence regarding newly weaned pigs. Transport may cause an acute stress response in weaned piglets, adding to the stress inflicted by weaning in itself. Handling during loading and unloading and mixing with unfamiliar individuals are important transport-associated stressors. A long duration of the transport may also cause dehydration and fastening periods. However, the fasting and water deprivation inflicted by transports less than 6–8 h is rarely detrimental to early weaned pigs; they are also fastening for long periods after weaning when they are weaned directly into a pen. The conditions of the transport seems to be important, and both season and space allowance affect the impact of transport. Weaned pigs are especially sensitive to heat stress during long transports, and cold temperatures must also be avoided. The reviews did not mention studies measuring the direct effect of transport on PWD.	21. Provide > 0.07 m^2^ of space per pig (Danish law requires >0.08 m^2^) 22. Ensure adequate thermoregulation during transport. 23. Review the handling, loading/unloading practices and design of the facilities; can stress be minimized? 24. Minimize mixing with unfamiliar animals before, during and after transport.	[46,47]

**Table 4 animals-12-01737-t004:** Management after weaning.

Item	What Do We Know?	Recommendations	Sources
Stocking density	High stocking density is a social stressor for the pigs after weaning, which leads to reduced performance and increased occurrence of disease. Studies measuring the direct effect of stocking density on PWD was not mentioned.	25. >0.34 m^2^ floor space per pig is recommended.	[7,29,44,48,49]
Group size per pen	Limited evidence indicated that there is no clear effect of group size on PWD incidence. Numerous studies on the effect of group size on post-weaning performance have been reviewed and meta-analyzed. A non-linear effect on growth rate and feed intake were present. A slight decrease in the growth rate and feed intake is seen when increasing the groups size from 10 up towards 100 pigs. However, keeping groups < 10 pigs yields clear improvements in growth rate and feed intake. Aggressive behavior after mixing does not seem to increase with increased group size.	No specific recommendations are given on this item.	[50,51,52]
Sorting of pigs into pens	Mixing of unfamiliar pigs is a social stressor to the pigs, it induces agonistic behaviors and increase the transmission of pathogens. This decreases the growth rate and feed intake, impairs the immune system and results in bite wounds and other negative outcomes. Thus, the indirect links to post-weaning gut health are clear; however, studies of the association to PWD incidence were not reported. Housing the pigs litter-wise prevent the above. There is not convincing evidence that pigs sorted by body weight performs better than haphazard sorting; in fact, the opposite might be the case. However, targeted feeding or wean-to-finish systems can be an argument for this practice. Sorting the pigs by sex has been reported to reduce fighting and aggressive behavior after mixing. Enrichment material can reduce the occurrence of aggressive behaviors. Co-mingling of piglets with unfamiliar individuals from other litters before weaning, preferably at 5–12 days of age, will enhance the social skills of the piglets and thus enhance their ability to cope with mixing after weaning. However, this will often only be practically feasible if new barn systems are built, and it may have other drawbacks (e.g., disease transmission).	26. Do not mix pigs originating from different sow herds. 27. Sort the pigs in litters, e.g., keep 2–3 full litters in pens with 30 pigs. 28. If mixing is unavoidable, sorting by sex might be beneficial. 29. If mixing is unavoidable, increasing the amount of enrichment material is another way to limit social stress from mixing. 30. Sorting the pigs by size might be a waste of labor; evaluate the benefit of the practice to justify its usage. 31. If using preliminary nursery pens just after weaning (e.g., “baby containers” or pens in the farrowing unit in a multisite system), preserve the pen composition rather than mixing pigs once again when moving the pigs into the permanent nursery pens.	[23,29,30,44,48,52]
Rooting and foraging material	If lacking rooting and foraging material, weaned pigs will perform different abnormal behaviors more frequently. One study was cited to show that provision of bedding material reduced the PWD incidence and increased the growth rate in weaned pigs. Based on the available evidence, we could not establish specific recommendations regarding what amount of enrichment is sufficient.	32. Provide rooting and foraging material (e.g., straw) in adequate amounts.	[41,53]
Room temperature and air flow	Cold room temperature and draught increase the incidence of PWD. Too high temperatures can reduce feed intake and growth rate. Fluctuations in the temperature decreases growth rate and increases the incidence of PWD.	33. If no focal heating (e.g., cover with heat lamps or cover with floor heating) is provided, the temperature should be 26–28 degrees C for the first 2 weeks after weaning. 34. Temperatures down to 23 degrees C is acceptable when focal heating sources are available (e.g., cover with heat lamps or cover with floor heating) 35. Ensure that the room temperature is stable and constant in the whole room. 36. Ensure that the pigs are not exposed to draught (cold air flow > 0.2 m/s).	[9,29,49,52,54,55]
Air quality	The concentration of dust particles, bacteria and toxins, ammonia, hydrogen sulfide gases and CO_2_ in the piggery have a substantial effect on post-weaning feed intake and growth performance. The association might be related to a marked effect on respiratory immunology and disease. An additional search was performed for reviews discussing the effect on gastrointestinal health without success; however, one original observational field study was identified. They found an association between high ventilation index score and low PWD occurrence. However, the index was a combination of measures of draught and gas concentrations. Good air quality may be ensured by adequate ventilation, thorough cleaning of the pens and by lowering the stocking density so that an air space (m^3^/pig) of at least 0.0118 × bodyweight (kg) + 1.82 is available (i.e., 1.9144 m^3^/pig for 8 kg pigs). Thresholds were not suggested for all the relevant substances in the reviewed literature; therefore, some of these were collected from a book chapter.	37. Ensure good air quality, i.e., NH_3_ concentration below 5 ppm, CO_2_ concentration below 1540 ppm, the concentrations of endotoxins below 1 μg/m^3^ and viable bacteria below 50,000 CFU/m^3^ air space and a concentration of dust below 3.7 mg/m^3^.	[29,49,54,56,57]
Weaning in farrowing pen	No literature was identified.	No specific recommendations are given on this item.	

**Table 5 animals-12-01737-t005:** Feeding and water strategy.

Item	What Do We Know?	Recommendations	Sources
Feeder space	Adequate feeder space minimize the competition for food and allows for group feeding, which meets the preferences of the piglets and facilitates social learning of eating. Pigs will start to eat earlier and generally increase the feed intake and growth performance when feeder space in increased. The risk of PWD has been reported to be higher in pig herds with limited feeder space. Based on the available evidence, we could not establish specific recommendations regarding the optimal feeder space/pig; however, we suggest that all pigs should be able to eat at the same time	38. Provide enough feeder space to allow all pigs within a pen to eat at the same time. 39. If recommendation 38 is not met, insert additional feeding troughs in the first 14 days after weaning to meet the recommendation in this risk period.	[7,9,52]
Feeding scheme	Feed can be provided either ad libitum or restricted by different feeding schemes. There is good evidence from both experimental and observational field studies that restricted feeding reduce the occurrence of PWD compared to ad libitum feeding. The weight gain will likely be reduced in the period with restricted feeding; however, this might be amended by a compensatory growth when switching to the ad libitum feeding later in the pigs life.	40. Practice restricted feeding for the first 14 days after weaning. 41. Provide 4–8 meals/day rather than, e.g., 1–2 meals/day.	[9,29]
Floor/mat feeding	Feed can be provided on the floor or on a mat in fully slatted pens. Limited evidence was cited regarding this practice and the cited review paper concluded that the available data was insufficient to document the efficiency of mat feeding. However, two studies had reported a decrease in mortality and removals for unspecified reasons. Mat feeding stimulates eating behavior the first day after weaning but also leads to increased feed wastage, especially after the third day post-weaning.	42. Provide feed on the floor for the first 1–3 days after weaning.	[52]
Wet/slurry/gruel feeding	Gruel feed can be prepared by mixing pellets or mesh feed with water (e.g., 1:1 or 1:2). Numerous studies documented that gruel fed pigs eat more and grow faster in the post-weaning period and that gruel feeding induce an increased small intestinal villus height and increase the abundance of lactic-acid producing bacteria in the intestinal tract. However, no cited studies reported on the effect on PWD occurrence, and one of the review papers concluded that the available data was insufficient to document the efficiency of gruel feeding. Gruel feeding requires labor. Providing a fraction of the full nutrient requirements as gruel in smaller meals, suggestively minimize feed wastage (as the whole meal is rapidly consumed) and ensures that the pigs get started with consuming the dry feed. Good hygiene must be kept in the troughs used for gruel feeding.	43. Supplement the dry feed with 3–4 small gruel meals/day for the first 7–14 days after weaning, while keeping good hygiene in the troughs.	[29,52,55]
Particle size	Grinding and hydrothermal processing of the feed results in fine particle sizes and thus increase the digestibility; however, this is clearly associated to stomach ulcers, and a coarser feed might be beneficial for the gut health. Weaned piglets prefer coarsely grinded diets. A limited amount of evidence suggests that provision of coarse feed to young pigs lowers the stomach pH, prolongs the stomach retention time, prevents ileal colonization of Salmonella and ETEC and increase the crypt depth and fermentation in the colon and thus the production of organic acids; i.e., coarse feed prevents factors involved in the causation of PWD. One study also reported inclusion of 4% coarsely grinded wheat bran reduced the PWD incidence in ETEC challenged pigs compared to challenged pigs fed feed with finely grinded wheat bran. Low-energy feedstuff can be used when adding coarse particles to minimize the loss from poor digestibility of the coarse particles.	44. Avoid particle sizes < 0.4 mm 45. Include particle sizes 0.5–1.6 mm 46. Include particle size > 1.6 while considering that it may mean a loss in digestibility. 47. Add large particles made from low-energy feed-stuff with digestible fibers.	[36,52,58,59,60]
Physical form of diet	Several studies document that the daily gain is higher for pelleted feed than mash. However, in pigs fed mash, the intestinal proliferation of Salmonella and *E. coli*/ETEC are lower feed than pigs fed pellets, and one study reported increased risk of PWD in pigs herds using pelleted diets. Young piglets prefer large pellets, and they find hard pellets difficult for to eat.	48. Use mash rather than pellets 49. If using pellets, use soft and large (e.g., Ø = 12 mm) pellets	[29,52,55,58]
Water accessibility	Liberal access to water will increase the water intake, which is associated with enhanced food intake and growth rate after weaning. The link to PWD incidence is not clear; however, good access to water is especially important for pigs suffering from diarrhea. Pigs prefer water sources located near the feeder rather than towards the alley or in the back of the pen. It takes a longer time for newly weaned pigs to learn drinking from nipples than from bowls, especially if the piglets are not familiar with nipples from their farrowing pen. They also spill less water when drinking from bowls. A too low rate of delivery may reduce the water intake.	50. Provide at least 1 water source per 10 piglets. 51. Insert an additional trough with water in the immediate post-weaning period, to meet the requirements in this risk period. 52. Bowls (with or without lever) is recommended over nipples for newly weaned pigs. 53. Place water sources near the feeder. 54. If not supplied in a bowl with large reservoir, water should be delivered at a rate > 0.45 L/min. 55. Water sources should also be available for pigs fed liquid feed.	[29,52,55,61]

**Table 6 animals-12-01737-t006:** Biosecurity and biomanagement.

Item	What Do We Know?	Recommendations	Sources
Cleaning before insertion	The load of pathogens and pollutants that activate the immune system in the barn environment can be dramatically reduced by cleaning the compartments properly before insertion of new pigs. Weaning into clean environments is associated with increased performance partly due to direct or indirect effects on gut health. A hygienogram (assessing the number of colony forming units in samples of the pen environment) can be used for quality assurance of the cleaning procedure.	56. Wash, disinfect and dry out the section before inserting a new batch.	[7,29,49,54,62]
All in/all out (AI/AO) practice	AI/AO practice entails inserting a batch of newly weaned piglets into a clean, empty room at the same time and allowing no movements or additional insertion of animals until the whole room is emptied again. This interrupts routes of disease transmission and thereby reduces the incidence of PWD (and other diseases) and generally increases the health and productivity of the pigs.	57. Practice AI/AO management strictly. 58. Consider that AI/AO routines might be violated by intermediate “baby pens” or supplementary insertion of pigs from nursing sows. Minimize the extent and impact of these breaches.	[7,29,49,63]
Personnel and their clothing acting as fomites	Stock personnel and other people visiting the farm can act as fomites and transmit infectious pathogens between sections, pens and pigs. Different measures can break the routes of transmission. Specifically, boots, clothing and hands/gloves are at high risk of being fomites.	59. Designated clothing should be available for different units of the herd. 60. Use and frequently change gloves and/or wash hands periodically. 61. Use foot baths between units: clean boots in preliminary baths using soapy water and a brush, then enter the disinfectant immersion for in adequate time according to producers instructions. Change the immersion preferably every day or at least every third day. 62. Alternatively: use different boots for each unit. 63. Suggestively, these measures generally recommended between different herd units could be applied specifically before entrance to sections accommodating newly weaned pigs.	[62]
Working routine from youngest to oldest animals	Newly weaned pigs will be more sensitive to certain pathogens than older pigs. The transmission of these pathogens can be prevented by implementing working routines where young pigs are not visited after contact with older pigs.	64. Visit and perform all necessary procedures in sections with the newly weaned pigs as the first thing in working day in the nursery unit. 65. If entrance is necessary later in the day, minimize the impact by adhering to the recommendations for *Personnel and their clothing acting as fomites* (see above). 66. Do not enter the nursery after visiting grower or finisher units.	[63]
Cleanliness of potential fomites	Tools and materials used in the herd might act as fomites. Movement of tools between sections may be prevented by having designated tools for each section, and these can be color-coded.	67. Make designated tools available for each section in the nursery. 68. Clean shared tools (e.g., herding boards), toys, troughs, etc. before using them in a new batch of piglets/another section.	[62,63]
Cleanliness of transport vehicles	In multisite farming systems, weaned piglets are transported from the farrowing unit to the nursery. The vehicles may be contaminated with pathogens from previous batches. Using the same vehicle for transportation of older animals (e.g., to the slaughterhouse) will pose an additional risk. Cleaning, drying and disinfecting vehicles hinders transmission.	69. Use (external) transporters adhering to certified cleaning schemes, e.g., transporters from the Specific Pathogen Free program in Denmark. 70. If using the farm’s own vehicle, enforce an appropriate cleaning and disinfection program between each batch. 71. Never use an uncleaned vehicle that has transported older animals.	[62,63]
Cleanliness of water source	Drinking water may be contaminated with pathogens. Either the water arriving from the plant or a local well can be of poor quality, or the systems within the herd can be contaminated with biofilms. In some regions (e.g., Denmark), the water is commonly delivered from public plants where the water quality is regularly checked. If this is not the case, or if a local well is used, the water quality should be checked at least annually. Poor water quality must be corrected.	72. Use water from water plants with quality assurance; 73. Or check the quality at least once a year yourself. 74. If the water quality is poor, correct the problem (using appropriate mechanical, physical or chemical treatments). 75. Water systems (tanks, pipes, etc.) in the herd must be cleaned and disinfected regularly.	[62,63]

**Table 7 animals-12-01737-t007:** Management of specific pathogens.

Item	What Do We Know?	Recommendations	Sources
Vaccination against ETEC	Currently, one live *E. coli* vaccine with F4ac and F18ac strains is commercially available in Denmark (COLIPROTEC^®^ F4/F18). The vaccine may be given orally either prior or after weaning. Finding the best time of administration may be difficult, as F18-receptors is first expressed from approximately 17 days of age and onwards. Additionally, lactogenic immunity may inhibit the effect of the vaccines in suckling pigs, while a weaning-induced acute stress response might impair the immunocompetence of the weaned pigs. Clinical effects of the vaccine can only be expected if it is timely administered and if the PWD is caused by ETEC strains matching the immunity induced by the vaccine.	76. Confirm the presence of ETEC with fimbria antigens matching the vaccine before considering vaccination. 77. Use the live oral vaccine, in case of ETEC-associated PWD. 78. Vaccinate pigs no earlier than 17 days of age. 79. Vaccinate pigs between 7 and 21 days before expected peak in ETEC-associated PWD incidence.	[64,65,66]
Vaccination against rotavirus	Vaccines against rotavirus are currently only marketed for (pre-farrowing immunization of) sows and the effect of this vaccine on PWD is unknown. Field trials of vaccines for neonate pigs have not consistently demonstrated good efficacy. The interaction with the lactogenic immunity, and the diversity of rotavirus groups and strains within groups makes it difficult to produce rotavirus vaccines that are efficient under field conditions.	No specific recommendations are given on this item.	[67]

## Data Availability

The data set from Study 1 will not be publicly available. The small community of veterinary pig practitioners in Denmark might make it possibly to identify the individual respondents based on the demographic characteristics they have provided.

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
