# Peer review of "Evidence-Based Recommendations for Herd Health Management of Porcine Post-Weaning Diarrhea"

_animals, 2022, doi:10.3390/ani12141737_

Round 1
Reviewer 1 Report
This is a very welcome paper on scientific relevant recommendations for herd health management of porcine post-weaning diarrhea. It is clearly written and gives a clear overview of measures that could support farm management advice. What I however miss is a short discussion on the outcome of this summary of measures and what the authors suggest as next step to implement this in practice. As Temple Grandin suggests, as cited in the introduction: ’successful transfer of knowledge and technology requires more work than doing the research.’ Could the authors include a short discussion and future view how this paper will contribute to adoption in the field? So how will the two parts in the paper link in the larger picture (the questionnaire and the summary of measures). When adding this short discussion, the readers are more taken by the hand for the larger picture and will maybe answer Temple Grandin’s concern.
Then, the paper could also win in value, when the authors give a short evaluation on the outcome of the relevant measures that were found. What I conclude when reading all recommendations is that the weaning procedure with all its elements is crucial in prevention and needs critical evaluation at farms. A shift in focus on the needs of the animals and supporting as much as possible seems to be the key. Although biosecurity still is mentioned and requires attention, this may not be the limiting factor at modern farms any longer? Maybe because biosecurity has been focus during the last decades. But maybe the authors have found other interesting links while summarizing all the measures that are worthwhile mentioning. The discussion as it is now is too short and leaves the readers a bit disappointed.
Author Response
Dear reviewer 1
Thank you for you critical revision. As you have requested, we have now added an additional discussion paragraph in line 254-368. We gratefull for you suggestion, as we think this have enhanched the quality of our work.
Best regards
Esben Eriksen
Reviewer 2 Report
The paper is very well written. Minor english/spelling/grammatical corrections required. Numbers in Figure 2 do not match (after subtracting one paper is missing) with those in lines 193-195. In Table 2 state what IUGR stands for the first time it appears.

Author Response
Dear reviewer
Thank you for reviewing our manuscript and for the very positive evaluation.
We regret to have bothered your with these simple typos and gramatical errors, and are very gratefull for the errors that were marked in the PDF. They have all been corrected.
We have corrected the typoe in figure 1 so it is in accordance with line 194.
We have written out the intrauterine growth restriction syndrome in the reccomendation 2 apperaring in table 2.
On behalf of the authors
Esben Eriksen
Reviewer 3 Report
The article „Evidence-based recommendations for herd health management of porcine post-weaning diarrhea“ (animals-1787600) describes the literature research for recommendations to prevent post-weaning diarrhea in pigs. Use of antibiotics will be reduced in the future and zinc oxide must be replaced by alternatives.
For decision making on farms the article shall support practitioners with evidence-based recommendations, which is of great benefit for decision making. The article is divided in two parts: in the first part practitioners were asked about their need for a decision-support tool. In the second part the evidence-based recommendations were developed.
Overall the recommendations given in the article are not new, so that they are well-known to swine practitioners. The value of the article is, that the reader can be sure, that recommendations are not only based on empirical knowledge, so that best practices are confirmed by this article. On the other hand several recommendations especially for alternative products, which can be used instead of antimicrobials and zinc oxide (as e.g. probiotics) are not mentioned, although they are often used in the field.
A negative list of alternatives, which have been examined for their beneficial effects would be also of use to save time and money for the farmer in the future. The authors excluded this topic from the beginning (line 87-89). Due to the fact, that this is a difficult task, because a lot of contradictory study results might be available in literature I can understand this decision. Nevertheless, at least in discussion a short paragraph about this topic (and the ambigious results and the reasons for them) should be added, at best together with personnel assessment of the authors (because they also decided on practical recommendations also based on their experiences (see figure 1 first box).
Line 65-67: The sentence about differing opinions about the best solution to solve the herd problem should be rephrased.
Line 108-110: Sentence is not clear: what was offered to the Danish pig practicioners ?
Line 128: Which period of time was covered by the literature research ?
Figure 2 “Papers prioritized n=81” Should this not be n=82 ? Please check.
Figure 2 Box “Papers not covering any of the items or outcome measures of interest”: Please specify the items which were not taken into account (describe in the running text).
Legend of figure 2: add number of reference.
Figure 3: Please describe in material and methods more details about the survey. Do the shown questions cover the whole part of the survey which was used for this study ? You can describe it in 2.1. What is meant in detail by the 5 questions shown in fig. 3 ?
E.g.: Is “assessment of occurrence of diarrhea” a tool ? What is meant by a “simple template for the planning of the process of outphasing of medicinal zinc” ?
Line 216-218: It is not clear which support-tools are meant
Line 219: What is meant by a “further innovation of this tool” ? Is it described somewhere ?
Line 202: The “validation of the concept” was not clear. Does it mean, that the “evidence based recommendations” were produced as a consequence of the answers of the survey ? This could be explained in more detail in the material and methods section.
Page 9 “gradual weaning”- ..”…however , the reviews suggesting it did so, did not cite any literature.” Please rephrase.
Table 1-7: please allocate the references to the different statements where possible
Line 248: Add a (short ?) discussion part about the described procedure and the value of recommendations. The recommendations are well known to the practitioners and farmers, but they are not implemented due to mainly non-scientific reasons (no time, no space, no personnel, different attitudes and beliefs).
Author Response
Dear reviewer 3
We are truely gratefull for your critical revision of our manuscript. We found your comments relevant and wise. We believe that our respones to your comments have enhanched the quality of our work. Pleas find a word document with specific response to your comments written in blue.
On behalf of the authors
Esben Eriksen
